# A scaling approach to estimate the age-dependent COVID-19 infection fatality ratio from incomplete data

**Beatriz Seoane** [ORCID] *

Departamento de Física Teórica, Universidad Complutense, Madrid, Spain

* beseoane@ucm.es

## Abstract

SARS-CoV-2 has disrupted the life of billions of people around the world since the first outbreak was officially declared in China at the beginning of 2020. Yet, important questions such as how deadly it is or its degree of spread within different countries remain unanswered. In this work, we exploit the 'universal' increase of the mortality rate with age observed in different countries since the beginning of their respective outbreaks, combined with the results of the antibody prevalence tests in the population of Spain, to unveil both unknowns. We test these results with an analogous antibody rate survey in the canton of Geneva, Switzerland, showing a good agreement. We also argue that the official number of deaths over 70 years old might be importantly underestimated in most of the countries, and we use the comparison between the official records with the number of deaths mentioning COVID-19 in the death certificates to quantify by how much. Using this information, we estimate the infection fatality ratio (IFR) for the different age segments and the fraction of the population infected in different countries assuming a uniform exposure to the virus in all age segments. We also give estimations for the non-uniform IFR using the sero-epidemiological results of Spain, showing a very similar increase of the fatality ratio with age. Only for Spain, we estimate the probability (if infected) of being identified as a case, being hospitalized or admitted in the intensive care units as function of age. In general, we observe a nearly exponential increase of the fatality ratio with age, which anticipates large differences in total IFR in countries with different demographic distributions, with numbers that range from 1.82% in Italy, to 0.62% in China or even 0.14% in middle Africa.

## 1 Introduction

The severe acute respiratory syndrome coronavirus 2 (SARS-CoV-2) has quickly spread around the world since its first notice in December of 2019. The pandemic of the disease caused by this virus, the coronavirus disease 2019 (COVID-19), at the moment of this writing, has claimed more than 400 thousand lives. Many countries in the world have declared different levels of population confinement measures to try to minimize the number of new infections and to prevent the collapse of their respective health systems. As the first wave of the outbreak

**Data Availability Statement:** This study uses third-party data. All the data analyzed has been extracted from public databases quoted in the manuscript (S1 Table). These data are also

collected and freely available at https://dc-covid.
site.ined.fr/fr/donnees/ and https://github.com/
datadista/datasets/tree/master/COVID%2019.

**Funding:** Beatriz Seoane was supported by the
Comunidad de Madrid and the Complutense
University of Madrid (Spain) through the Atracción
de Talento program (Ref. 2019-T1/TIC-12776) and
partially supported by Ministerio de Economía,
Industria y Competitividad (MINECO) (Spain)
through Grant PGC2018-094684-B-C21 (also
partly funded by the EU through the FEDER
program).

**Competing interests:** The authors have declared
that no competing interests exist.

starts to be controlled, the question of how to proceed next arises. The daily number of deaths is progressively decreasing in Europe, and with it, the majority of the countries are starting to release the national lock-downs. The design of future strategies will be sustained on the evolution of the official statistics, and the problem is that these statistics are very defective and incomplete. This is so because, on the one hand, the total number official cases is strongly limited by each country's screening capacity, which means that only a small fraction of the total infections is correctly identified (typically those presenting symptoms above a certain level of severity fixed by each country's policy). On the other hand, the shortage of screening tests and an overwhelmed health system also tend to underestimate the number of deaths in the official records. The actual degree of under-counting for both measures is unknown and most likely country dependent, combined with the fact that the pandemic is still on going, results in largely irreconcilable case fatality ratios (CFR) all over the world [1–3].

Efforts have been made to determine the clinical severity of the virus [4–8] and its dependence with factors such as age [9], sex [10] or comorbidities [11–13], but determining precisely how deadly this virus is remains hard [14, 15]. Many different solutions using the available data have been proposed to extract the correct CFR [2, 16–21], estimate the number of infections [22, 23] or the infection fatality ratio [24–27]. Even the results of some early sero-epidemiological tests sampling the population degree of immunity have been strongly controversial [28, 29]. Probably the most reliable estimations for the infection fatality ratio (IFR, the probability of dying once infected) as a function of the patient's age, were proposed by Verity *et al.* in Ref. [25] using the data from 4999 individual cases in mainland China and exported cases outside China. The ratios obtained were further validated with the reported cases in the Diamond Princess cruise ship [30]. Yet, these estimations were based on two assumptions. Firstly, a perfect detection of all the infections among people in their fifties, a debatable hypothesis given the difficulty of systematically identifying all the mild and asymptomatic infections. And second, that the virus had spread uniformly within the population of all ages, which is rather improbable because they were analyzing mainly infections among travelers (that tend to be younger). Nevertheless, the picture is clear, the lethality of the virus increases sharply with the patients' age, being particularly deadly for elderly people and mild for kids.

In the absence of a reliable number of confirmed infections, most of the statistics have focused on the number of deaths, which are expected to be a fraction of the first one. But deaths are much less common than infections, which means that in order to estimate correctly the infections, one needs a very accurate death counting. In this sense, it is widely accepted that the number of real deaths linked to COVID-19 is noticeably larger than what officials statistics say [31, 32], but estimating precisely how much is hard and will likely depend strongly on the country data collection policy and capacity. One can try to estimate the size of this discrepancy from the excess mortality observed since the beginning of the pandemic in the public death records. This approach, though apparently infallible, is not without difficulties. Indeed, in most of the countries, the epidemic peak took place at the same time as that of the lockdown measures, which means that, on the one hand, the mortality for accidents and injuries has decreased [33–36], and on the other hand, the health system being under a lot of stress, the mortality linked to lack of medical assistance for other diseases has strongly increased too [37]. Correcting these effects in the reference mortality trend requires a careful an exclusive analysis.

## 2 Materials and methods

In this work, we attempt to estimate the IFR as function of age using scaling arguments relating the cumulative number of deaths reported in different countries and age groups. We provide

**Table 1. Estimations assuming a uniform attack rate.** We show our estimation for the uniform infection fatality rate (UIFR) before and after quantifying the effects of the systematic under-counting of deaths. We also estimate the percentage of the population infected in each country by the end of May of 2020. Errors include the statistical error (±*sigma*, the standard deviation obtained through error propagation of the results in Table 1, and the uncertainty of the prevalence survey in Spain) and a systematic error of 35% of possible under-counting of deaths, see Section 4.2).

| | Uniform infection fatality rate | | % Population infected | |
|---|---|---|---|---|
| age group | with under-counting | estimation without under-counting | Country | |
| 0-9 | 0.0012%(4) | 0.00118%(0.00082-0.0016) | Spain | 5.0%(4) |
| 10-19 | 0.0021%(7) | 0.00211%(0.0014-0.0028) | Portugal | 1.0%(4) |
| 20-29 | 0.009%(23) | 0.00878%(0.0065-0.012) | Norway | 0.33%(12) |
| 30-39 | 0.024%(5) | 0.0241%(0.019-0.032) | Korea | 0.06%(2) |
| 40-49 | 0.072%(18) | 0.0722%(0.056-0.097) | Italy | 4.3%(16) |
| 50-59 | 0.26%(5) | 0.256%(0.21-0.35) | Germany | 0.8%(3) |
| 60-69 | 0.84%(0.14) | 0.839%(0.71-1.1) | France | 3.4%(12) |
| 70-79 | 2.8%(5) | 3.47%(2.9-4.7) | England | 6%(2) |
| 80-89 | 8.9%(18) | 12.7%(11.-17.) | Denmark | 0.9%(3) |
| 90+ | 23.%(7) | 42.1%(34.-57.) | | |

all the details concerning the databases used for the analysis in the Materials section below (Section 2.1) and the definitions of our variables in the Methods Section (Section 2.2). We then use these age-distributed measures to establish a direct correspondence between the mortality rates in patients below 70 years old (where we argue the official counting is more accurate, see Section 3.1) published in different countries around the world (but mostly in Europe) in Section 3.2. This good correspondence allows us to make predictions about the degree of spread of the virus in different populations, or the global IFR of a country, as compared to another one. We also observe that the collapse of the mortality rate with age in different countries is compatible with a pure exponential increase of the IFR with age (assuming a uniform attack rate). The scale of total infections is then consistently fixed from the rate of immunity obtained via blood tests of a statistical sampling of the citizens Spain in Section 3.3 (and compared to seroprevalence tests in Geneva, Switzerland, and New York City, United States). This scale allows us to compute the IFR as function of age and the number of current infections in each country that are given in Table 1. In addition, we estimate the probability of being detected as official case, needing hospitalization and intensive care (if infected) as function of age in Spain in Section 3.4. All these rates are obtained under the assumption of a uniform attack rate, an assumption that seems fairly reasonable seeing the immunity measures of the Spanish test, measures that, when once taken into account, do not change qualitatively the results discussed so far (see in Section 4.1). Finally, we estimate the extent of the under-counting of deaths linked to COVID-19 among the elderly in the different countries (assuming, again, a uniform attack rate) and give estimations for the overall lethality of the virus in Section 4.2.

## 2.1 Materials

We provide below the details and sources concerning the data used in the analysis.

**2.1.1 Age profile of the COVID-19 deaths.** We study the distribution of cumulative deaths by age-groups in different countries and regions. In general, we consider only COVID-19 confirmed deaths (that of patients tested positive for the disease). In order to quantify the possible under-counting of deaths associated to COVID-19, we also consider registers of the deaths were COVID-19 appeared in the death certificate, even as a simple suspicion, details are given in the Under-reporting of deaths subsection.

*National data*. The information about the distribution of the number deaths associated to COVID-19 with age in different countries is taken from the database prepared by the "Institut national d'études démographiques (Ined)" (France) freely available for scientific use at the website https:/dc-covid.site.ined.fr/fr/donnees/. For the rest of epidemic's measures in Spain (cases, hospitalizations, entries to intensive care unit and deaths), we used the COVID-19 data-dista database [38]. In both cases, these databases collect together the official information published by each country's health authorities. More details about each country's data sources and apparition of these data in the paper are given in S1 Table.

Some countries give the age profile for a sub-group of the total number of deaths. If this were the case, we assumed a uniform sampling of the ages in all the age segments, and we renormalize all the cumulative deaths by age so that the sum of the deaths over all the age groups matches the total number of deaths published by each country on the 22nd of May of 2020.

**Regional and local data** In addition to the national data, we also discuss the age-profiles of different regions in France, Switzerland and Unite States of America in Section 3.3. For the distribution of COVID-19 deaths with age by department in France, we used the data furnished by Santé Publique France, in particular the "donnees-hospitalieres-classe-age" available at the Données hospitalières relatives à l'épidémie de COVID-19 website (data downloaded the 20/05/2020). The information about the COVID-19 deaths in the Canton of Geneva is taken from the "N. 5—18 au 24 mai 2020" report in the République et canton de Genève website. The information about the deaths in New York city is taken from the "Total Deaths" reports of NYC health website,

*Under-reporting of deaths*. We estimate the degree of under-reporting of deaths linked to COVID-19 by comparing systematically the number of deaths having COVID-19 mentioned in their death certificate (even if the link was just a mere suspicion), with the number of deaths having laboratory confirmation for COVID-19. In order to compare data between age groups, we normalize this difference by the number of confirmed deaths, that is:

$$\text{Fraction of under-counting} = \frac{\text{Deaths (suspected \& confirmed)-Deaths (confirmed)}}{\text{Deaths (confirmed)}}. \quad (1)$$

The data concerning deaths mentioning COVID-19 in the death certificate was taken from the "up to week ending the 22nd of May" report in the ONS website (England and Wales) and the "Informe de situación 22 de mayo 2020" from Comunidad de Madrid website.

The age distribution of the official data (to generate Fig 2) is taken for (England only) from the Ined database (which is extracted from the daily report of the National Health Service that includes only deaths tested positive for Covid-19 occurred in hospitals only). In order to account for the deaths in Wales, we multiplied the English distribution by 1.05 (Wales deaths represent a 5% of the sum of the deaths of Wales and England in the ONS report). In order to estimate the official age distribution of deaths in Madrid, we renormalized the national age distribution of cumulative deaths by the official cumulative number of Madrid at the 14th and 22nd of May. This is a reasonable approximation considering that almost a third of the total COVID-19 deaths in Spain occurred in Madrid.

**2.1.2 Demographics information.** For the demographics distribution of the different countries, we used the data available at the Ined database which corresponds to the last distribution published by each country official statistics' agencies (more details can be found in S1 Table), and the database from the "World Population Prospects" of the United Nations https:// population.un.org/wpp/Download/Standard/Population/ (the estimation for 2020) for the discussions about demography distribution in other parts of the world and their expected effect in the Global IFR (see Section 4.3). The demographics of the Geneva canton was extracted

from Statistiques cantonales in the République et canton de Genève website. For the demographics of New York City we used the data published in the NYCdata website from 2016.

## 2.2 Methods

Statistical offices and health institutions of many countries have been publishing regularly the age distribution of the cumulative number of deaths occurred in their territory since the beginning of the outbreak. We have combined national data from Denmark, England & Wales, France, Germany, Italy, South Korea, Netherlands, Norway, Portugal and Spain, regional data from Geneva (Switzerland) and Madrid (Spain), and local data from New York City (Unite States of America). Unless something else is mentioned, we consider 10 age groups, each gathering together patients with ages in the same decade (with the exception of the patients over 90 years old, which are grouped together). Since the different age segments are not uniformly populated, and this distribution changes significantly from one country to another, we discuss always the number of deaths normalized by the number density of people $x_\alpha(\mathcal{C})$ in each age group $\alpha$ and country $\mathcal{C}$, that is,

$$\hat{D}_\alpha(t;\mathcal{C}) \equiv \frac{D_\alpha(t;\mathcal{C})}{x_\alpha(\mathcal{C})}, \tag{2}$$

being $D_\alpha(t;\mathcal{C})$ the cumulative number of deaths at a time $t$. In the following, we will refer to $\hat{D}_\alpha(t;\mathcal{C})$ as the *normalized* cumulative number of deaths and we will omit the country variable $\mathcal{C}$, unless explicitly needed. We show in Fig 1A the evolution of $\hat{D}_\alpha(t)$ in France for our ten age groups. As shown, once the effects of the demographic pyramid are removed (the fact that there are much more people in their fifties than in the nineties in any population, for example), the mortality expands over almost five orders of magnitude between kids and elderly people.

Asymptotically, that is, for a large number of total infections in a country, the cumulative number of deaths in each $\alpha$ at a given $t$, will be a fixed fraction of the cumulative number of infected individuals in that group, $I_\alpha$, at a previous date $t - \Delta$, thus $\Delta$ is an effective time related to the time elapsed between infection and death (estimated to be, in average, around 20 days [39–41]). In general, $\Delta$ depends on $\alpha$ and on the country $\mathcal{C}$, but we omit it here for simplicity because the differences are extremely subtle at this time of the outbreak. Then,

$$D_\alpha(t) = f_\alpha I_\alpha(t - \Delta) + \mathcal{O}\left(\sqrt{f_\alpha I_\alpha}\right), \tag{3}$$

being the proportionality factor, $f_\alpha$, the infection fatality ratio (IFR) for the age group. Note that the probability that $n$ people of age within $\alpha$ die (among a total number of infections $I$), is described by a Binomial distribution, $B(n, p)$, where $p$ is the probability of being infected and dying at that age (i.e. $p = I_\alpha f_\alpha / I$). Then, the expected number of deaths, is $E(n) = Np = f_\alpha I_\alpha$ and the expected error of this value is $\mathrm{Desv}(n) = \sqrt{Ip(1-p)} \approx \sqrt{f_\alpha I_\alpha}$ since $p \ll 1$. The assignment of a unique delay for all the cases is, of course, an over simplification, but which yet works quite well as the number of infections becomes large. We show, for instance, the perfect match in time between the cumulative number of cases and deaths at a later time in Spain in S1 Fig.

In general, we do not know either the total number of infections $I = \sum_\alpha I_\alpha$, or the number of infections in a particular age segment $I_\alpha$, but we know the latter should be an (essentially constant) fraction of the total number of infections, plus fluctuations, that is,

$$I_\alpha(t) = r_\alpha x_\alpha I(t) + \mathcal{O}\left(\sqrt{r_\alpha x_\alpha I}\right), \tag{4}$$

 

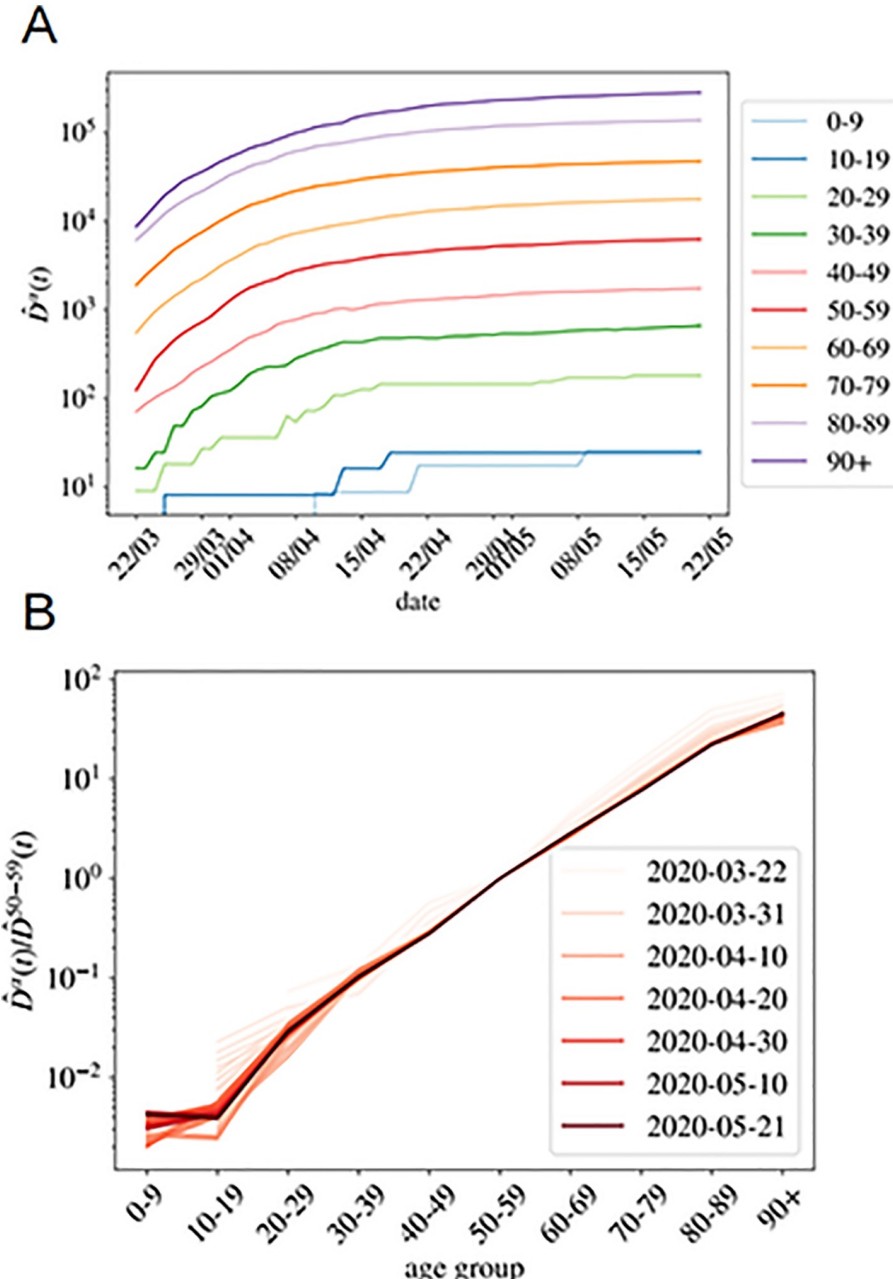

**Fig 1. Normalized number of deaths occurred in French hospitals as a function of age. A** We show the evolution with time of the cumulative number of deaths normalized by the number density of individuals in age group $\alpha$ (i.e. $\hat{D}_\alpha(t)$ in Eq (2)). In **B**, we show $\hat{D}_\alpha(t)/\hat{D}_{50-59}(t)$ as function of the age group, for all the times in **A** (the darker the color, the more recent the measurement, and we give some dates in the legend). This quotient is essentially time-independent as discussed in Eq (7), and it lets us estimate the quotient between the UIFR (the IFR under the assumption of uniform attack rate, see Eq (6)) of the two age groups, that is, $\hat{f}_\alpha/\hat{f}_{50-59}$.

with $r_\alpha(\sim I_\alpha/x_\alpha I)$ being the relative risk of infection for group $\alpha$ as compared to the probability of infection if all ages had the same probability of getting infected (that is, $r_\alpha = 1$ for all $\alpha$). In other words, $r_\alpha > 1$ (or $r_\alpha < 1$) means that group $\alpha$ is more (or less) prone to being infected than by random. Note also, that, by definition, $\sum_\alpha r_\alpha x_\alpha = 1$. This $r_\alpha$ has the advantage of being

 

dimensionless, and differs from the standard definition of attack rate for an age-group, which would be $r_\alpha I/N$, with $N$ the total country population. Recent results analyzing the spread of the virus within close contacts in the outbreak in China suggests a uniform exposure of the virus across the population [42], meaning that $r_\alpha = 1$ for all the groups (quite different from the patterns observed for the seasonal flu [43, 44]). There is, however, an important debate whether the low fatality observed in patients below 20 years old is related to a low risk of death or a low risk of infection. For the moment we keep this variable free and we will discuss it at the end of the paper. This risk of infection $r_\alpha$ could, in principle, vary with time, but we do not observe a systematic change with time (at least in the period studied). This will be clearer with the discussion around Fig 1B for the cumulative deaths, or for the analogous figure concerning the daily measures (which should be more sensitive to a change in $r_\alpha$) in S2 Fig.

## 3 Results

### 3.1 The counting of deaths is more accurate below 70 years old

The under-counting of deaths comes from mainly two sources: (i) only the deaths that can be directly linked to COVID-19 (by means of a positive result in a PCR test, typically) are included in the official counting and (ii) countries mostly count the deaths occurred within hospital facilities in the statistics. Source (i) tells us that all the patients that die before being tested are invisible. This will happen eventually at all ages but since old patients are more prone to develop severe symptoms and have more difficulties to seek immediate medical attention, this situation will be far more common among the elderly. Also source (ii) mainly affects old people because being hospitals crowded, the oldest patients have been often treated in retirement/care homes or in their own homes. For these reasons, we expect a significantly more accurate reporting of the deaths of younger patients (in particular, under 70 years old). As we show below, it is also possible to quantify this idea.

According to the Office of National Statistics in the United Kingdom, among deaths mentioning COVID-19 in the death certificate (in England and Wales by the 22nd of May) 64% took place in hospital, 29% in care houses and 5% at home [45]. Analogous data published by the Community of Madrid's government (which counts more than 1/3 of the official deaths in Spain) reports similar ratios: 61% hospitals, 32% socio-sanitary places and 6% home. France counts separately the deaths occurring in hospitals and in care homes, and the latter being almost 60% of the former. Deaths occurring in care houses are a large portion of the total in all countries, which means that an incomplete counting there, modifies notably the overall statistics. However, once we look at the mortality per age group, such under-counting only affects the patients of a certain age. In fact, we can compare the number of deaths having COVID-19 mentioned in the death certificate (even if it is only a suspicion, which most probably represents an over-counting of the real deaths) and the official counting of deaths linked to COVID-19. In Fig 2, we show fraction of under-counted deaths with respect to the official numbers (see the definition in Eq (1)) for England and Wales and the Community of Madrid. In both places, the under-counting is relatively age independent under 70-80 years old, and very important above, specially for the patients above 90 years old, where real numbers may probably double the official counting. Furthermore, this mismatch is getting worse as records in England and Wales are correctly updated (in Madrid it seems rather stabilized). Details on the data used to generate these plots are given in the Materials.

In summary, we expect a small mismatch between the real and the official number of deaths among patients under 70 years old (the $\sim 30\%$ of under-counting is probably too large because deaths caused by other diseases are probably also included in this count), and a much higher systematic under-counting for the older segments. The actual numbers will depend on the

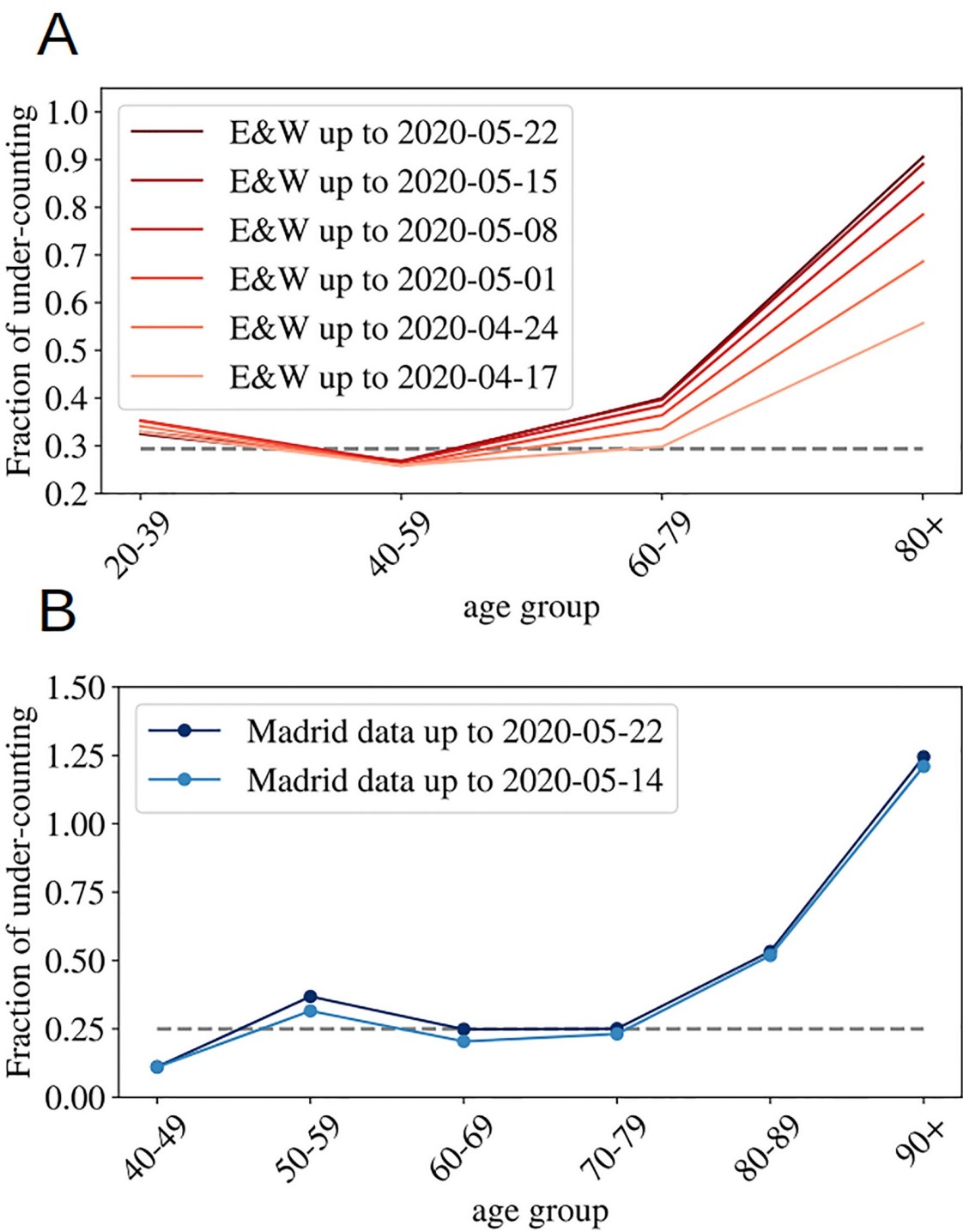

**Fig 2. Under-counting of deaths per age groups.** We show the fraction of under-counted deaths, per age groups, observed when comparing the number of deaths certificates where COVID-19 was mentioned either confirmed or suspected, and the official deaths attributed to COVID-19, relatively to this second number, see Eq (1) for the definition, for England and Wales in **A**, and for the Community of Madrid **B**. The horizontal lines mark the mean rate of 'under-counting' below 80 years old.

country capacity to detect quickly the infections, but also on the particular details concerning the counting of official deaths (which establishments are considered). We give these details, together with the last date used for each country in the Methods and Dataset section.

## 3.2 Scaling between age segments

The combination of Eqs (3) and (4) tells us that:

$$\hat{D}_\alpha(t) = \hat{f}_\alpha I(t - \Delta) + \mathcal{O}\left(\sqrt{\hat{f}_\alpha I/x_\alpha}\right), \tag{5}$$

where

$$\hat{f}_\alpha = r_\alpha f_\alpha, \tag{6}$$

would be the probability of dying with age $\alpha$ if the virus attacked uniformly all ages within the population. In other words, this is the "apparent" fatality since it weights how deadly the virus is (statistically) for a patient in an age group, with the relative risk of getting infected at that particular age. For this reason, we refer to $\hat{f}_\alpha$ as the *uniform* infection fatality rate (UIFR) (i.e. the IFR under the assumption of uniform attack rate between ages), as compared to $f_\alpha$, which is the *real* (potentially non-uniform) IFR associated to the disease. Both measures are only equal if $r_\alpha = 1$ for all $\alpha$.

All together, for all age segments, $\hat{D}_\alpha(t; c)$ is expected to be proportional to the total number of infections at a previous date, $I(t - \Delta)$. Alternatively, the quotient between the mortality rate of two distinct age groups,

$$\frac{\hat{D}_\alpha(t)}{\hat{D}^\beta(t)} = \frac{\hat{f}_\alpha}{\hat{f}^\beta} + \mathcal{O}\left(\sqrt{\hat{f}_\alpha I/x_\alpha}\right) + \mathcal{O}\left(\sqrt{\hat{f}^\beta I/x^\beta}\right) \tag{7}$$

should be time independent (as long as the number of the expected deaths for each group is large enough), and equal to the quotient between the UIFR of each group. This is precisely what we observe for the deaths occurred in French hospitals (see Fig 1B) where we show the quotient between each $\hat{D}_\alpha(t)$, and the deaths among patients in their fifties, $\hat{D}^{50-59}(t)$ for all daily reports since the 22nd of March of 2020 (the darker the color the more recent the measurements). The other countries considered shows qualitatively the same behavior, we decided to show France because it has been reporting age statistics (on a daily basis) for the entire number of deaths occurred up to that date. Thus, with this kind of analysis, even if we do not know the exact mortality associated to the virus, we can determine how deadlier it is, at least apparently, for an age group as compared to another. We say apparent, because up to here, we cannot distinguish if the virus seems less aggressive for an age segment because the lethality is low (that is, $f_\alpha \ll 1$) or because so few individuals of that age got infected (that is, $r_\alpha \ll 1$).

The same kind of arguments applies to data from different countries at a fixed time. Indeed, one expects that the IFR, $f_\alpha$ should not vary too much from country to country (at least within countries with comparable health systems). However, the relative attack risk $r_\alpha$ may do. Yet, if these differences are not large, also $\hat{f}_\alpha$ should be country independent. In such case, Eq (5) tells us that the different $\hat{D}_\alpha(\mathcal{C})$, essentially differ by a multiplicative constant proportional to the total number of infections, $I(\mathcal{C})$, in each country. We show in Fig 3A, the counting $\hat{D}_\alpha$ by the 22nd of May of 2020 available for the different countries where we found information about the death profiles by decades of age (see Materials section for details) as a function of $\alpha$.

As argued, the different countries' curves are essentially parallel in logarithmic scale, with the exception of the Netherlands, where the mortality increases in the elderly segments must

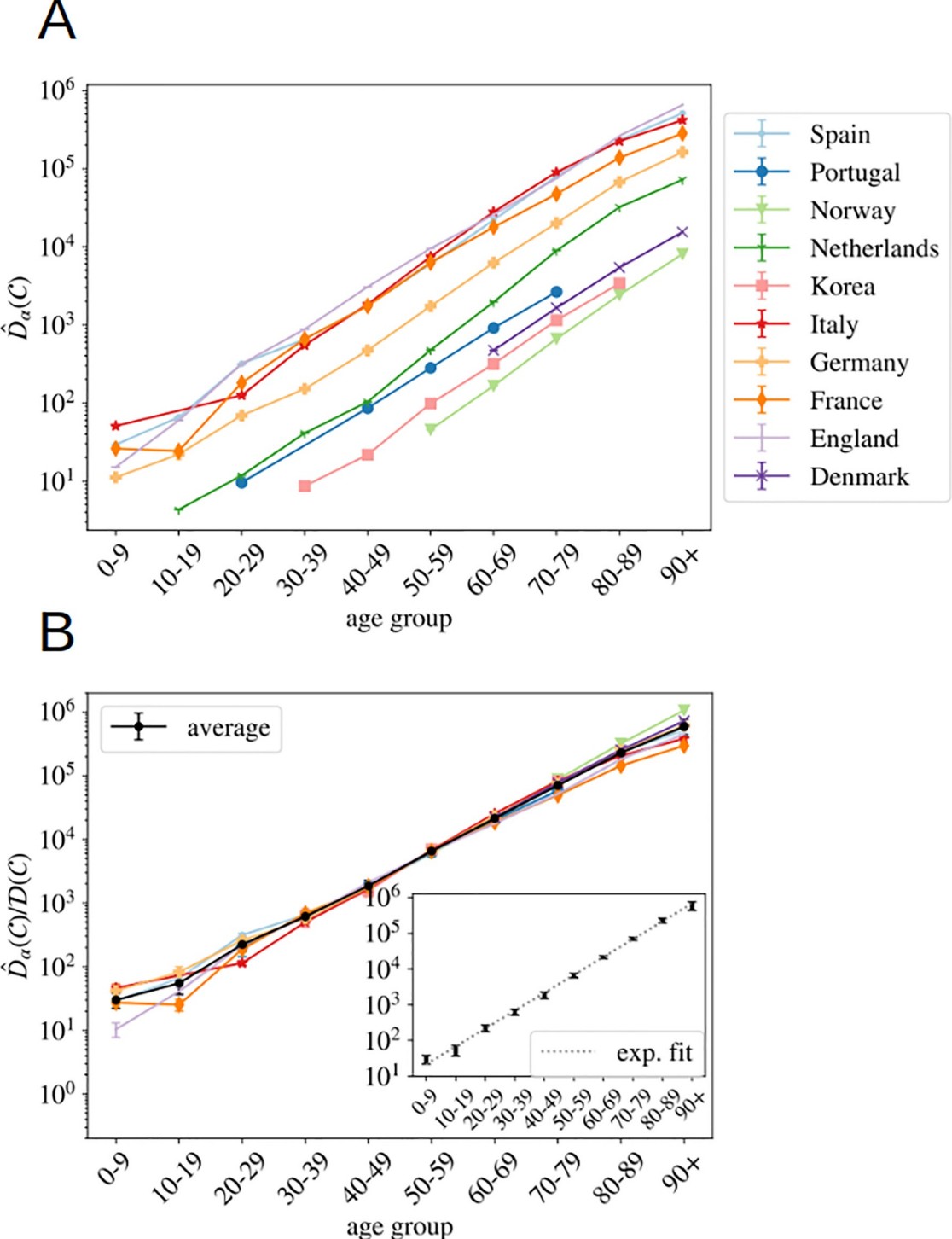

**Fig 3. Normalized number of deaths in different countries as a function of age. A** We show the normalized number of deaths per age group (defined in Eq (2)) for a selection of countries affected by the COVID-19 epidemic at very different scales. In **B**, we show the same data (excluding the Netherlands) but where each country has been multiplied by a constant $\mathcal{D}(\mathcal{C})$ so that it collapses with the Spanish curve in the age region in between 30 and 70 years old. The values of each country's constants are given in S2 Table. In black, we show the country average for each age segment (errors calculated with the boostrap method up to a 95% of confidence), and in **C** the fit of this average to a pure exponential function, see Eq (8).

faster than the rest of the countries (we do not known the reason, it might be related to a significantly different $r_\alpha$). In other words, we can extract both the number of total infections and the UIFR by age (but for a multiplicative constant common to all the countries, or all the ages, respectively) from the collapse of these curves. We show in Fig 3B this collapse (where Netherlands was excluded even if the curve collapses well with the rest below 70 years old), which works extremely well for all the countries in the age region between 30-69 years old (despite the different orders of magnitude of $\hat{D}_\alpha(\mathcal{C})$). Deaths below 30 are very rare, which means that strong fluctuations between countries are expected (see Eq (5)). The collapse is less satisfying above 70 years old, but, as discussed, we believe it is mostly related to a different degree of under-counting of deaths for these segments of age (though other effects, such as an effective protection of the elderly population might be an important effect in some countries too). Yet, we believe that it is mostly related to under-reporting effects, because, for instance, the French curve would quickly match the rest of the countries if one added (for the segment over 80 years old) the official deaths occurring in care houses to the hospital deaths shown here (see for instance, the data published by Sante Publique France https://www.santepubliquefrance.fr/maladies-et-traumatismes/maladies-et-infections-respiratoires/infection-a-coronavirus/articles/infection-au-nouveau-coronavirus-sars-cov-2-covid-19-france-et-monde). We will try to estimate the extent of this under-reporting in each country below.

One can now exploit this similarity in the increase of cumulative deaths with age between countries to remove the statistical fluctuations. Thus, the country average of this collapse gives us the UIFR (but for an unknown proportionality constant $\hat{f}_0$ common to all age segments). We give the values of this average in S2 Table. Data obtained is compatible with an exponential growth of the UIFR with age (as shown in Fig 3C). In fact, we obtain a very good fit of the data to

$$\bar{\hat{f}}_\alpha \propto \exp\left(\mathcal{A} \times \text{age}_\alpha\right) \tag{8}$$

with $\mathcal{A} = 0.115(7)$. In particular, we used the least squares method to fit $\log \bar{\hat{f}}_\alpha$ (and its error) as function of $\text{age}_\alpha$ via a linear regression. The good quality of the fit is evaluated through the low value of the $\chi^2$/d.o.f = 3.8/8. This strong dependence of the fatality with age anticipates a widely variable global UIFR ($\sum_\alpha x_\alpha \hat{f}_\alpha$) between countries due to the different demographic distributions. We will discuss this point in Section 4.3. Let us stress that we show this fit with a purely descriptive purpose, since we shall not use these results any further in the analysis.

Furthermore, the collapsing constant is essentially the relative of the total number of infected people in a country with respect to our reference country, that is, $I(\mathcal{C})/I(\text{Spain})$. This is not entirely true due to the different country policies concerning the death-counting, but, as discussed, we estimated that the unreported fraction under 70 years old is inferior to 30% (see Fig 2) and the quotient of the under-estimation of the two countries would, in general, much smaller. We show these collapsing constants in S2 Table.

### 3.3 Fixing the scale

**3.3.1 Number of infections and uniform fatality rate.** Up to this point, we have only obtained the number of infections by country with respect to the number of total infections in Spain, and a quotient proportional to the UIFR (the IFR assuming uniform attack rate) by age. In both cases, the proportionality constants (though both related) are unknown. In order to fix the scale, one can look at the statistical studies of prevalence of antibodies against SARS-Cov2 in different populations. In particular, we refer to the preliminary results of the sero-epidemiological study of the Spanish population (inferred from 60983 participants) made public by the

Spanish Health Ministry the 13th of May of 2020 [46], that estimates that only a 5.0% (95% interval of confidence (IC): 4.7%-5.4%) of the Spanish population had been infected (from blood tests drawn in between 27/04-11/05/2020). Also, as an independent control of the scale, we use the results of an analogous sero-prevalence survey of the residents of the Geneva, Switzerland (from 1335 participants) [47].

The sampled rate of immunity in the Spanish population allows us to fix $I$(Spain) in S2 Table and with it, estimate the number of infections in each of the countries shown in Fig 3, as summarized in Table 1). The results obtained are lower, but compatible, with the independent estimations by Phipps et al. [23] or Salje et al. for France [48], and compatible with the results of small antibody prevalence survey in England [6.78% (95% C.I. 5.21%-8.64%)] [49] and marginally compatible with a survey among blood donors in Denmark [1.7% (95% C.I. 0.9%-2.3%)] [50]. As shown, the rates of infection (for the entire country) are rather low, in particular compared to the 60-70% herd immunity threshold (even if it were lowered for other effects [51]). Yet, it is important to stress that the propagation of the virus has been rather heterogeneous in the territory, being the contagion rather high in certain regions and insignificant in others. We take for example France, where the age distribution of the COVID-19 deaths is available for all the departments (see Materials). Using also the data up to the 22nd of May, we estimate that the percentage of the population infected has reached 12% in the Island of France (the department of Paris), 7% in the Great East, 2.5% in Upper France, and it is 1% or less in the rest of departments.

Furthermore, the total number of infections allows us to estimate the UIFR as function of the age in Spain just by dividing our $\hat{D}_\alpha$ by this number, that is, using Eq (3),

$$\hat{f}_\alpha(\text{Spain}) \sim \frac{\hat{D}_\alpha(\text{Spain})}{I(\text{Spain})}. \tag{9}$$

We show the values obtained using this formula in Fig 4A. Then, we can extract $\hat{f}_0$ from the comparison of $\hat{f}_\alpha$(Spain) with the values $\hat{f}_0\hat{f}_\alpha$ in Table 1, in the age regions where we believe that the counting of deaths is reliable (the region where the collapse of Fig 3B is good). We use the group 50-59 to fix this constant ($\hat{f}_0 = \hat{f}^{50-59}_{\text{Spain}}/\hat{f}^{50-59}$), which allows us to reconstruct entirely our estimate for the averaged UIFR (we show these values in Fig 4A and Table 1). This determination of the UIFR is expected to underestimate the fatality ratio for the oldest segments of population, we will try to correct this bias in the next section. We will also include this corrected estimation in Table 1).

We can test the accuracy of the estimated IFRs by this method, using another independent sero-epidemiological survey. In particular, we use the work by Stringhini *et al.* [47] that measures the degree of seroprevalence in the canton of Geneva (Switzerland) from samples of 1335 participants. Up to the 24th of May of 2020, the canton's authorities had reported 277 deaths, all but one in patients above 50 years old. We can use the age distribution of these deaths and our estimation of the IFR in Table 1, to guess the fraction of the population that have been infected so far using Eq (3). We show in Fig 4B, the quotient $D_\alpha/x_\alpha\hat{f}_\alpha N$, being $N$ the total population of the canton of Geneva. If our $\hat{f}_\alpha$ were, indeed a good estimation for the real IFR, this quotient should give us the fraction of the population infected in that age group, which was estimated to be very similar above 50 years old and equal to 3.7% (95% CI 0.99-6.0) and about 8.5% (95%CI 4.99-11.7) in between 20-49 years old [47]. As shown, our predictions are in very good agreement with the survey estimation (specially once the systematic under-counting of deaths in the estimation of the IFR is corrected, see Section 4.2).

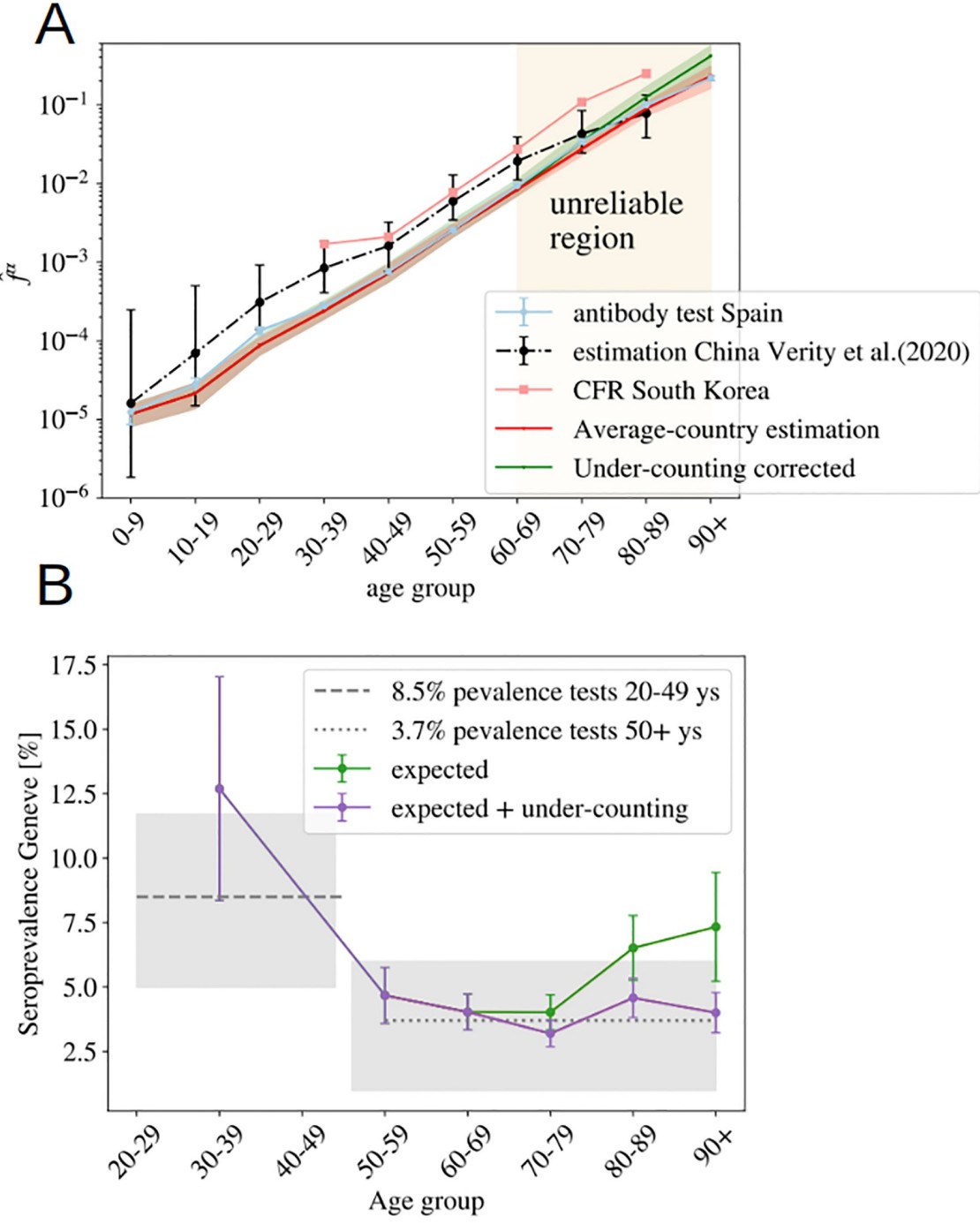

**Fig 4. Probabilities assuming a uniform attack rate. A** We use the measurements of the number of infections in Spain to estimate the UIFR using Eq (3) in both regions. We fix the constant $\hat{f}_0$ in S2 Table using the estimation of the UIFR in Spain for the age group 50-59 to infer the values of the country average UIFR (from the collapse of Fig 3B). We show this first estimation in red, and in green, we show the UIFR after correcting the under-counting of deaths over 70 years old. We compare these results with the estimation by Verity *et al.* [25] and the CFR (i.e. the probability of dying for confirmed COVID-19 cases, not the IFR) by age in South Korea. In **B**, we use the IFR estimations from A and Table 1, to predict the seroprevalence of anti-SARS-CoV-2 antibodies in the population of Geneva, Switzerland, from the official distribution of deaths per age of a total of 277 deceases. The predicted fraction of infections is given in dots (in green, if we used the bare estimation of Eq (9), in violet, if we include the corrections linked to under-counting). In horizontal lines (and the 95% of confidence interval in gray shadow), we show the actual values measured from the antibody survey of Ref. [47] in patients of different age-groups.

The perfect match between the results in Spain and Switzerland (and in a lesser detail with England and Denmark) lends great confidence to the estimated ratio between deaths and infections. Yet, let us stress that these estimations might be only valid for similar health systems, similar percentages of comorbidities in the population, and for hospitals not too overwhelmed during the worst moments of the epidemic peak. In fact, if we use the IFR of Table 1 to estimate the percentage of infections in New York City (NYC) from the distribution of the deaths by age published by NYC Health at different dates (we show the results in S3 Fig), we obtain predictions for the overall antibody prevalence that evolve in time from 27% (data from 15th of April), 48% (the 1st of May), 57% (the 15th of May), to 63% (the 2nd of June). In other words, this would suggest that herd immunity would had already been reached in the city. However, there are proofs that this is not true. Indeed, the presence of antibodies within the NYC's citizens was randomly sampled during the last weeks of April, in the base of a survey of 15000 people in all the New York State. The results announced by the Governor in a press conference the 2nd of May of 2020 reported that only a 19.9% of the tested presented antibodies. If we move forward ∼ 20 days in time to see this reflected in the deaths [39, 40], we overestimate the infections by a factor 3, which inevitably suggests that the IFR was higher in New York City that what it was in Spain or in Geneva, unless there are issues in the sero-prevalence study, something hard to estimate because technical details of the survey have not been published so far (to our knowledge). The origin of this mismatch might be multiple: a non universal access to health care, higher presence of comorbidities among the young population and/or collapse of hospitals. For this point, we would like to stress that the effects of a possible sanitary collapse must be more evident in NYC than nowhere else, given the disproportionate dimension of the NYC outbreak with respect to the rest of countries considered here. For instance, just in NYC there were almost twice more deaths in patients below 50 years old than in the whole Italy during the Spring of 2020.

We can also compare our IFR with previous estimations. Our numbers are smaller than the estimation by Verity *et al.* [25] for all the age segments except those that concern the elderly patients (though still compatible with their confidence interval for most of the age groups), and about three times smaller than the CFR (the probability of dying among the confirmed cases) per age group measured in South Korea (where a massive number of screening tests were made). This difference could be explained, in both cases, from an under-estimation of the total number of infections. On the one hand, the IFR in Ref. [25] was estimated from the CFR, and the statistical prevalence of antibodies among the travelers returning home from repatriation flights (which represents a much lower sampling that the one considered in the Spanish survey). On the other hand, Korea has been very successful identifying new infections by tracking the social contacts of the infected, but it is very unlikely that they are able to trace all the infections.

Before ending this Section, we want to warn about the limitations of the current sero-epidemological surveys, which will probably affect our results (even though we would like to stress that the Spanish survey has been praised for its robustness [52]). In fact, extracting accurate results from them is challenging for different reasons. Firstly, because the study must be well designed to avoid undesirable bias in the recruitment of the participants. Secondly, because the probability of detecting the antibodies change with time [53] (an effect that must be taken into account [54]). Thirdly, because available tests are not very accurate [55], which means that statistical adjustments must be included in the analysis to avoid mistaking the antibody rate with the false positive rate [56]. And finally, because the spread of the virus have been very heterogeneous in space (as we illustrated for France above), which means that very large samples are necessary to get the correct picture of a country.

### 3.4 Other probabilities

Spain also gives age distributed data (for groups of patients with ages in the same decades) for the cumulative number of official cases, $C_\alpha$, new hospitalizations, $H_\alpha$, and new admissions in intensive care units, $S_\alpha$. Due to the shortage of screening tests, for most of the age groups, the number of cases gives us a measure of the number of patients with symptoms severe enough to visit an emergency room. For the oldest groups, it might not be the case because care houses with confirmed cases have been more systematically tested than the rest of the population. Then, we apply the same reasoning used to compute the UIFR to these indicators, which allows us to estimate the probability of being included in each of the other three categories (always assuming uniform attack rate). Unlike the deaths, policies concerning who get tested, hospitalized and/or admitted in an intensive care unit probably depend strongly on the country, which means that these probabilities might not be directly extrapolated to other countries.

Eq (5) reads for a general observable $X$ ($X = C$, $H$, $S$, or $D$),

$$\hat{X}_\alpha(t) = \hat{f}_\alpha^X I(t - \Delta_X) + \mathcal{O}\left(\sqrt{\hat{f}_\alpha^X I/x_\alpha}\right), \tag{10}$$

which means that we can directly extract the probability of being included in the $X$ category $\hat{f}^X$ using the measure $I$(Spain) from the antibody prevalence study [46]. Note that knowing the precise value of $\Delta_X$ is not crucial here because the propagation of the disease was essentially interrupted in Spain during by the end of May, which means that $I(t - \Delta_X)$ changes very little with time at this point. We show the estimations of these probabilities per age group in Fig 5.

We see that, between 20-80 years old, the probability of being confirmed as a case does not depend too much on age, and it keeps fixed around 1 every 10 infections. The probability is higher for older segments and much smaller for people below 20 years old. For the other indicators, we observe a strong dependence of all levels of severity with age. For the intensive care

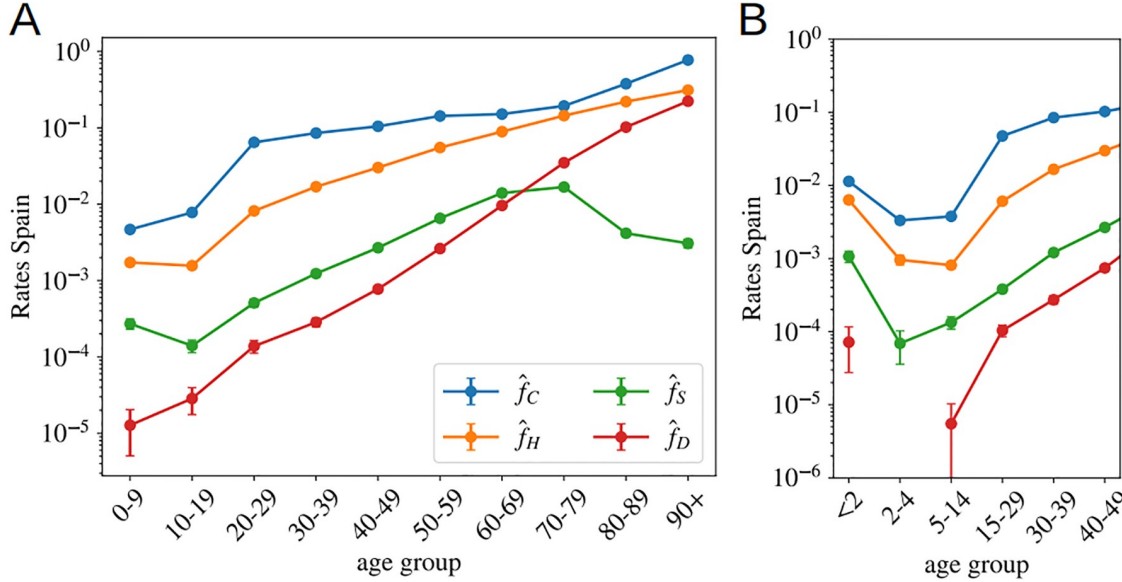

**Fig 5. Other probabilities as function of age assuming uniform attack rate.** In **A** we show the probability of being classified as official case, $\hat{f}_C$, being hospitalized, $\hat{f}_H$, admitted in intensive care, $\hat{f}_S$, and dying, $\hat{f}_D$, in Spain in Spring 2020, as function of the age using age segments of 10 years. **B**, we show the same data but were the kid's information has been grouped by smaller age-segments, evidencing the severity of the cases in patients under 2 years old. **A** is generated using the data by the Spanish Health Ministry up to the 22nd of May and **B** with the data published by the RENAVE, see Materials and S1 Table.

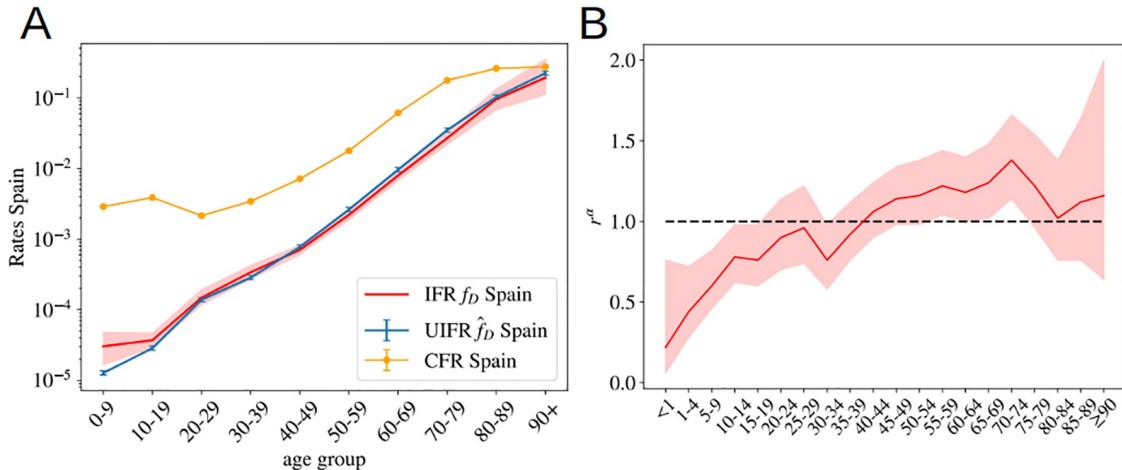

**Fig 6. Uniform versus non uniform IFR. A** We show the relative risk of infection for an age segment $r_\alpha$ (see Eq (4) and definition below) taken from the sero-epidemiological study of the Spanish population [46]. While the youngest segments of the population seem to be less hit by the virus, the distribution of the infections is rather similar to that of a uniform attack rate, indicated by the dashed line $r_\alpha = 1$ here. The 95% confidence interval for $r_\alpha$ is indicated by the red shadow. **B** We show the estimated uniform and nonuniform IFR for Spain and compare it with the CFR as a function of age. The error for the non-uniform IFR is shown by a red shadow.

unit admissions, however, above 70 years old, one sees clearly the effects of the policies regulating the access to intensive care with age, an access that becomes rare over 80 years old. A situation which certainly contributes to increasing slightly the mortality rate for the oldest age groups. We show in Fig 5B narrower age groups concerning the youngest patients. This second Figure tells us that the severity related to COVID-19 in children is rather heterogeneous in age, being particularly dangerous for kids below 2 years old (an age segment for which the admissions in intensive care are more common than for patients above 40 years old as shown in Fig 5B). Furthermore, these probabilities might be underestimated by the uniform attack rate assumption, since one expects a significantly lower exposure to the virus at these low ages (we will see this confirmed in the data shown in Fig 6).

## 4 Discussion

### 4.1 On the non-uniform distribution of infections

Our indicator for the IFR, the UIFR $\hat{f}_\alpha$ (and the probabilities of presenting different degrees of acuteness), measure how more probable is to die with a given age, which is not necessarily the true IFR (that is, the probability of dying once infected, our $f_\alpha$ in (3)). The two observables are only equal if the contagion is uniform among all age segments of the population (we recall that, in our definition, $f_\alpha = \hat{f}_\alpha / r_\alpha$, and uniform attack rate implies $r_\alpha = 1$). In other words, with our approach we are not able to distinguish if the mortality is low in a particular age segment because (i) the disease is mild at these ages (low $f_\alpha$) or (ii) because this age segment is rarely infected (low exposure, $r_\alpha \ll 1$ in Eq (4)). Previous studies estimating the IFR per age group, for instance Ref. [25], assumed a uniform spread of the virus, something that seems justified by contagion dynamics studies [42].

The sero-epidemiological study [46], gives also some clues about this point, because it also estimates the attack rate for different age groups. We can extract our relative risk, $r_\alpha$, from the estimated attack rate (we recall that the attack rate given by $r_\alpha I/N$, with $N$ the country-population). We show the values we obtain in Fig 6A. The measures only report a significantly lower

spread among children (which might be related to the closure of the schools during the lockdown), but for the rest of the ages the distribution is not so far away from the uniform attack rate. In any case, no exponentially increasing attack rate with age is found to balance the strong increase of the fatality with age. However, the much lower exposure of the kids to the virus tells us that the probabilities estimated in Fig 5B might be underestimated in that age segment, something that could change the overall picture of the severity of COVID-19 in babies, that might be similar to that of the adults. The change of tendency of the severity with age in the case of infants could related with the suspected connection between the COVID-19 and Kawasaki diseases [57–59].

We can nevertheless compute the real (non-uniform) IFR using these values for $r_\alpha$ for the Spanish data, and compare it with our previous estimation. We show the results in Fig 6B. As shown, both estimations are essentially compatible for all the age segments, which lends confidence to our previous results. The real fatalities will slightly change once the effect of the non-uniform attack rate is included, but we do not expect these non-uniform fatalities to change drastically with respect to the uniform estimations we gave above.

## 4.2 On the under-counting of deaths

As discussed above, one expects the number of deaths associated to COVID-19 to be underestimated in the official statistics, specially on what concerns to the elderly people. In this section, we try to estimate by how much. The collapse of Fig 3B shows us that Norway reports a higher number of deaths in the age segments above 70 years old than the rest of the countries, while the scaling of the normalized number of deaths in lower age groups are fairly similar to other countries. We believe that their counting is more accurate than in the rest of countries for two reasons. Firstly, because the Norwegian authorities reported deaths (of patients tested positive for COVID-19) occurring everywhere: hospitals (38%), caring and retirement houses (59%) and homes (2%). And second, because the country was much less affected than the rest of countries considered (Norway has reported only 235 deaths so far), which means that they are much better equipped to properly detect and treat all the infections. For this reason, we can use the Norwegian measures to estimate quantitatively our under-determination of the IFR among the elderly. In particular, we estimate an under-estimation of the mortality in the elderly groups of 70-79: 22%, 80-89: 40% and 90+: 86%. We show in Fig 4B, that this simple (and uncorrelated) correction allows us to predict correctly the measured prevalence of antibodies among the oldest people in the canton of Geneva (Switzerland) [47].

Yet, from the comparison with the Norwegian data we can only argue in terms of the scaling of the IFR of an age segment with respect to other, but not on the factor common to all age segments. For this, we can use the comparison between our estimation for the UIFR based on official COVID-19 deaths and those where COVID-19 appeared mentioned in the death certificate. The sero-prevalence study [46] estimated that a 11.3% of the population of the Community of Madrid had been infected, so we can use this number to estimate the IFR of the region. Such a IFR has to be regarded as an upper limit of the real one, because "suspicion of COVID-19" probably encompasses many other respiratory diseases. We show this IFR compared to our previous estimation, and the estimation after correcting the under-counting of the oldest segments (using the Norwegian death data) in S4 Fig. We observe that, firstly, the "Norwegian" correction introduced for the elderly segments is in perfect agreement with the scaling observed in the Madrid regional data, with attaches confidence to this correction, and second, that Madrid's estimation is around 35% larger than our previous estimation for all age groups. This comparison gives us an upper limit of the real IFR, which means that it allows us to estimate the maximum error of the predictions given up to now (as discuss, the real IFR is

**Table 2. Country-dependent estimates.** We estimate the percentage of unreported number of deaths for each country together with the expected fatality ratio once included these estimated missing deaths. In the parenthesis we include the expected values if the current death counting was perfect (no missing deaths, left side of the parenthesis) and if heavy under-counting was present, such as the one observed when comparing with number of deaths with COVID-19 in the death certificate (right side of the parenthesis).

|  | % of missing deaths | % total IFR |
|---|---|---|
| Spain | 38.%(0-86) | 1.6%(1.1-2.1) |
| Portugal | 9.1%(0-47) | 1.3%(1.2-1.8) |
| Norway | 0%(0-33) | 1.2%(1.2-1.6) |
| Korea | 16.%(0-57) | 0.87%(0.70-1.2) |
| Italy | 61.%(0-120) | 1.8%(0.98-2.4) |
| Germany | 32.%(0-78) | 1.6%(1.1-2.1) |
| France* | 110%(0-190) | 1.6%(0.84-2.2) |
| England | 79.%(0-140) | 1.3%(0.88-1.8) |
| Denmark | 29.%(0-74) | 1.3%(0.97-1.7) |

*France numbers were computed using only the deaths occurring in hospital facilities, which means that a 58% of under-counting is already confirmed with the counting of deaths occurring in care-houses. We cannot correct the minimum IFR because we do not have the age profile of these deaths.

expected to lie in between the estimation based on the official COVID-19 deaths and this suspected deaths' one). We show these estimations in Table 1 and Fig 4A after taking the effects of under-counting into account.

We can use these corrections to estimate the number of unreported deaths for each of the countries considered and the values of the UIFR per age to compute the global IFR of each country. We show this data in Table 2. Considering that a lower diffusion of the virus among the elderly would result also in a lower apparent mortality in these groups, we give also the expected total IFR if the actual counting were perfect (left side of the parenthesis), and if a constant 35% of under-counting was present in all the age groups (right-side of the parenthesis).

## 4.3 On the overall infection fatality ratios and demographics

The values of Table 2 shows us that the global fatality of the disease depends strongly on the demographics pyramid of each country, which is a direct consequence of the nearly exponential dependence of the UIFR with age. In fact, we can use the average values given in Table 1 to explore how the global IFR would change in different parts of the world just due to a different distribution of the number of citizens with age (that is, leaving aside the differences related to the different health systems or economical/social conditions). This observation was previously proposed in [60]. With our estimations, we expect that, while for Italy the IFR would be 1.8%, the same IFR age profile predicts a 0.62% IFR in China (extremely similar to the one estimated in Ref. [25]) or a 0.14% in middle Africa, which could explain, partially, why the outbreaks have been significantly less important there than in Europe (where the overall IFR would be 1.38%).

## 5 Conclusions

We have studied the scaling of the cumulative number of deaths related to COVID-19 with age in different countries. After normalizing these numbers by the fraction of people with that age over the entire population, we observe that the lethality of the disease increases (almost) exponentially with age, expanding over almost 5 orders of magnitude between the 0-9 and 90+ age segments. In addition, we show that this scaling with age is essentially country independent for ages under 70 years old. We argue that the differences observed over this age are

mostly related to different levels of under-counting of deaths among elderly people. The collapse of the mortality data allows us establish direct correspondences between the cumulative number of infections occurred in each country since the beginning of the outbreak.

At a second stage, we use the Spanish survey of the sero-prevalence anti-SARS-CoV-2 antibodies in the Spanish population [46] to fix the scale between the number of infections and the number of deaths, which allows us to estimate the COVID-19 infection fatality ratio as function of age (under the assumption of uniform attack rate). We evaluate these numbers with an analogous prevalence survey of the Genova canton [47]. We also show that, when applied to the COVID-19 death profile of New York City, our predictions are not compatible with the antibody rates estimated by the New York State [61]. This observation suggests that either the real immunity rate is much higher (and reached herd immunity levels) or the fatality ratio has been significantly higher in New York City than in Spain or Geneva, a discrepancy that might be related to a different health system, a higher prevalence of comorbidities in their population or a collapse of the sanitary system during the worse moments of the epidemics. The scale of the number of infections allows us to compute as well the probability (if infected) of being classified a case, hospitalized, admitted in intensive care units or dying in Spain. The results show a clear increase of all degrees of severity with age, with the notable exception of the infections in patients below 2 years old that lead to much more complications than for older young patients, a situation that could be aggravated by the low exposure of this population to the virus during the lock-down measures.

We further discuss the validity of the uniform attack rate hypothesis using the age distribution of the antibody rates in the Spanish sero-epidemiological study, concluding that even if differences of exposure of the virus between ages are observed, differences do not change qualitatively our estimations for the infection fatality ratio. However, the low attack rate measured among babies warns us that our estimations for the infection fatality rate below 2 years old might be importantly underestimated.

We use information concerning the number of death certificates where COVID-19 was referred as possible death cause to show that the under-counting of deaths is a problem that mostly concerns the deaths of old patients. We use the scaling of the mortality with age in Norway to estimate the real fatality ratio of the elderly age segments (in other words, reverse the under-counting). We then test these estimations with the age profile of deaths in the canton of Geneva and of the deaths certificates in the Community of Madrid.

Finally, our analysis relies exclusively on public statics' data and can easily be updated as more accurate information is available (for instance regarding the attack rates in different countries or better estimations of the total number of infections). For instance, severity rates are now known to be strongly dependent on the patients sex [10] or comorbidities [13] too, features that could be directly included in this analysis with no effort and that would greatly help to understand the interplay between them and age. In addition, if consolidated, the probabilities and the approach explained here, can be easily used to estimate the degree of penetration of the SARS-CoV-2 in different cities, regions, or countries, and to track the evolution of the pandemics.

Finally, but not least, we want to stress that we only analyzed the changes of the total mortality with age, but the socio-economical environment of the patients plays also an important role. This study could be generalized to include such variables.

## Supporting information

**S1 Table. Source and details of the age-distributed data (by country) used in the analysis.** In the last column, we detail the Figures and Tables generated with these data. All these data are freely available for scientific use at the INED's website: https://dc-covid.site.ined.fr/fr/

(PDF)

**S2 Table. Collapse of the mortality rate in different countries.** We give the values extracted from the collapse of Fig 3B: the increase of the mortality with age (proportional to the uniform fatality ratio $\hat{f}_\alpha$) and the number of infections in each country with respect to the number of infections in Spain $I_C/I_{\text{Spain}}$ equal to the collapsing constant $\mathcal{D}(\mathcal{C})$. The relative scaling of the mortality above 70 years old is expected to be significantly underestimated. Errors are obtained using the boostrap method at 95% of confidence. The errors of $\mathcal{D}(\mathcal{C})$ are only the statistical errors extracted from the data collapse, they do not include the systematic error associated to the different policies of death counting the different countries which would be much larger, we try to give a better estimate below.
(PDF)

**S1 Fig. Simple scaling relation linking the evolution of the cumulative number of deaths and the cases with time.** We show the evolution with time of the cumulative total number of official COVID-19 cases and deaths in Spain. In the inset the deaths' curve is displayed 5 days backwards in time and multiplied by 9, following very precisely the cases' evolution once it surpassed approximately the 100 cases. Please not that cases are confirmed much later than the infection date and later than the onset of apparition of symptoms.
(TIF)

**S2 Fig. Daily normalized number of deaths registered in French hospitals as function of age and time. A** We show the daily measures of deaths for age group $\alpha$ (normalized by the population density at this group), $\Delta \hat{D}_\alpha(t)$, for different dates. The darker the color, the more recent the measure. In **B** we show the collapse of the data when we normalize the data with the numbers of group 60-69 years old. Distinct date data collapse worse in a single curve than in the case of the cumulative number of deaths in Fig 1 because being the daily measures smaller, the fluctuations are much larger, yest, we do not observe any systematic change of the attack risk $r_\alpha$ with time.
(TIF)

**S3 Fig. Predictions for the sero-prevalence in New York City.** We show our predictions for the sero-prevalence presence in New York City using the death age profile published at different dates and the IFR of Table 1 (without under-counting corrections). Our predictions are significantly higher than the results of the sero-epidemiological survey announced by the New York State Governor the 2nd of May of 2020.
(TIF)

**S4 Fig. Estimation of the uniform infection fatality rate by age for the Community of Madrid using the number of deaths where COVID-19 was mentioned in the death certificate (black dots), compared with our estimation of the UIFR extracted from the average of several countries (blue line) and the same estimation where the fatality of the oldest segments was adjusted to take into account the systematic under-counting of elderly deaths (estimated using the Norwegian distribution of deaths with age).** We see that this correction match very well the scaling observed in Madrid's data.
(TIF)

**S5 Fig.**
(PDF)

## Acknowledgments

I would like to thank Aurélien Decelle, Luca Leuzzi, Enzo Marinari, Giorgio Parisi, Federico Ricci-Tersenghi, Riccardo Spezia and Francesco Zamponi for useful and interesting discussions, and to Elisabeth Agoritsas, Ada Altieri, Alessio Andronico, Marco Baity-Jesi and David Yllanes for a critical and constructive read of the manuscript.

## Author Contributions

**Data curation:** Beatriz Seoane.

**Formal analysis:** Beatriz Seoane.

**Methodology:** Beatriz Seoane.

**Resources:** Beatriz Seoane.

**Software:** Beatriz Seoane.

**Validation:** Beatriz Seoane.

**Visualization:** Beatriz Seoane.

**Writing – original draft:** Beatriz Seoane.

**Writing – review & editing:** Beatriz Seoane.

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
