## [Decision Letter · Decision Letter 0]

16 Sep 2020

PONE-D-20-17835

A scaling approach to estimate the COVID-19 infection fatality ratio from incomplete data

PLOS ONE

Dear Dr. Seoane,

Thank you for submitting your manuscript to PLOS ONE. After careful consideration, we feel that it has merit but does not fully meet PLOS ONE’s publication criteria as it currently stands. Therefore, we invite you to submit a revised version of the manuscript that addresses the points raised during the review process.

Your manuscript was reviewed by 2  experts in the field. Both found many important problems in your submission and produced copious comments. Please carefully review the attached reviews and provide point-by-point responses.

We look forward to receiving your revised manuscript.

Kind regards,

Yury E Khudyakov, PhD

Academic Editor

PLOS ONE

Journal Requirements:

'The funders had no role in study design, data collection and analysis, decision to publish, or preparation of the manuscript.'

Reviewers' comments:

Reviewer's Responses to Questions

**Comments to the Author**

1. Is the manuscript technically sound, and do the data support the conclusions?

Reviewer #1: Partly

Reviewer #2: Yes

2. Has the statistical analysis been performed appropriately and rigorously? 

Reviewer #1: I Don't Know

Reviewer #2: Yes

3. Have the authors made all data underlying the findings in their manuscript fully available?

Reviewer #1: Yes

Reviewer #2: Yes

4. Is the manuscript presented in an intelligible fashion and written in standard English?

Reviewer #1: No

Reviewer #2: Yes

5. Review Comments to the Author

Reviewer #1: Review of "A scaling approach to estimate the COVID-19 infection fatality ratio from incomplete data"

This paper studies age-patterns of death due to COVID-19. The goal is to try to identify regularities across countries; these regularities can then be used as the basis for predicting unobserved quantities in some countries based on the experience of other countries where more complete data are available. For example, a high-quality sero-survey is available in Spain, and the paper tries to develop methods can can be used to project information from the Spanish sero-survey onto data from other settings, where no sero-survey is available.

This is an interesting and important topic, and it seems clear that the analysis described in the paper is based on a thoughtful consideration of many aspects of this problem. However, the paper currently reads like a very early draft, to the point where it is quite difficult to understand what is happening in the analysis. There are some minor issues with the English language, but that is not the primary concern -- rather, the organization, lack of key detail, and inclusion of what seem like extraneous details, are the primary challenges I had in understanding the argument here. I think it's possible that the ideas here could be the basis of a paper that is publishable, but in my opinion, this draft needs major revisions before it is ready to really be evaluated.

Major comments:

* pg 2, 2nd paragraph - I find the assertion that mortality due to non-COVID causes has decreased to be surprising. Is that really known definitively? For many causes, it's plausible that death rates have increased: routine medical care has slowed down, lots of people are likely suffering from depression and other mental health challenges because of the economic and social disruption. If this claim is made, there should be citations or evidence provided for it. Otherwise, it seems like speculation

* pg 2, I would find it helpful if the information about the data sources was summarized in a table

* pg 3, I don't think that eqn (1) is the mortality rate (or death rate) as conventionally used in demography/epidemiology, because the denominator does not appear to have anything related to time. (Conventionally, a mortality rate has in its denominator a measure of exposure, ie people X time). For example, unless I'm reading eqn (1) incorrectly, it says that a population observed for twice as long -- but with no change in mortality conditions -- would have twice as high a value of \\had{D}_\\alpha. This may not cause problems for the rest of the analysis, but anyone used to working with demographic/epidemiological data would likely find it confusing to call it a 'mortality rate'. So, assuming the lack of time in the denominator is intentional, I suggest calling this something like 'cumulative fraction dead' or, anyway, something other than 'mortality rate'

* pg 3, 'expected error of an un-normalized histogram' - what does this mean? Where does this come from? There is no citation or explanation. I don't think I've ever seen this phrase in an epidemiological paper before. This is likely related to my confusion about what is being treated asymptotically (see 'additional comments')

* pg 4, is r_\\alpha not net of population size? It looks like r_\\alpha = \\frac{I_\\alpha}{x_\\alpha I}; that seems counter-intuitive; the first time I read this, I assumed the natural quantity to look at would be r_\\alpha = \\frac{I_\\alpha N}{x_alpha I}. So I would find it helpful to motivate and explain in more detail what r_\\alpha is

* Fig 2 - how is 'excess deaths' defined in the calculations shown in this figure? It looks like a relative quantity, but I'm guessing - there's no detail provided about this. Pg 5 says 'we give these details ... in the Methods and Dataset section' -- but I don't see any section with that title?

* pg 5 - "'apparent' fatality (what we perceive from the daily news)" - this sounds like a useful point, but I did not really understand how \\hat{f}_\\alpha is what is shown in the news. I suggest explaining this in greater detail

* Pg 6-7 / Fig 3 - How exactly was this fit performed? Maximum likelihood? Least squares? Also, I wasn't convinced that the efforts to assess model fit took into account the complexities of the log-log scale, which can make assessing fit difficult. On that topic, I find that this paper has a useful discussion:

Clauset, Aaron, Cosma Rohilla Shalizi, and Mark EJ Newman. "Power-law distributions in empirical data." SIAM review 51.4 (2009): 661-703.

(The authors may have done this kind of thing - but it's not described in the paper.)

Additional comments:

Note: there are many typos, missing words, and other writing issues throughout the paper. I will note some of them here, but I eventually stopped keeping track of these. The paper needs a lot of editing.

- abstract: I find it a little confusing to refer to the "growth" of the mortality rate with age; we don't usually talk about death rates 'growing'. I would suggest "increase" instead of "growth"

- abstract: 'validate' seems like a strong word for the analysis here; perhaps 'evaluate'?

- abstract: I believe the IFR is the 'infection fatality ratio', not 'fatality infection ratio'

* pg 2, 1st paragraph - infection fatality ratio is defined, but not case fatality ratio; I suggest defining CFR the same way IFR is defined, in parenthesis after first using the term

* pg 2, 1st paragraph - 'cruiser' is not the right word. "Cruise ship", maybe?

* pg 2, 1st paragraph - it's not clear to me what is meant by "given how elusive the detection of the virus is". Is this referring to a lack of testing? Or the difficulty of devising accurate tests?

* pg 2, 3rd paragraph -- 'dimension' does not seem like the right word here. Maybe the 'extent' of under-counting of deaths?

* pg 3, 'Asymptotically' - I can't understand what 'asymptotically' refers to without setting up what is getting arbitrarily big. Is it the sample size? The amount of time elapsed? Is either of those scenarios important here for some reason?

* pg 3, 'assignation' - I don't think this is the right word here

* Fig 1, caption - what is UIFR?

Reviewer #2: When you state the sentence "The actual degree of under-counting for both measures is unknown and most likely country dependent, which results in largely irreconcilable case fatality ratios all over the world." you should back it with a citation (Chirico F, Nucera G, Magnavita N. Estimating case fatality ratio during COVID-19 epidemics: Pitfalls and alternatives. J Infect Dev Ctries. 2020;14(5):438-439. Published 2020 May 31. doi:10.3855/jidc.12787).

In Introduction you should move this "We define all our variables in Section 2.2. We establish a

direct correspondence between the mortality rates in patients below 70 years old (where we argue the official counting

is more accurate) published in different countries around the world (but mostly in Europe) in Section 3.2. This good

correspondence allows us to make predictions about the degree of spread of the virus in different populations, or the

global IFR of a country, as compared to another one. We also observe that the collapse of the mortality rate with age

in different countries is compatible with a pure exponential growth of the IFR with age (assuming a uniform attack

rate). The scale of total infections is then consistently fixed from the rate of immunity obtained via blood tests of a

statistical sampling of the citizens Spain in Section 3.3 (and compared to seroprevalence tests in Geneva, Switzerland,

and New York City, United States). This scale allows us to compute the IFR as function of age and the number of

current infections in each country that are given in Table 2. In addition, we estimate the probability of being detected as

official case, needing hospitalization and intensive care (if infected) as function of age in Spain in Section 3.4. All these

rates are obtained under the assumption of a uniform attack rate, an assumption that seems fairly reasonable seeing the immunity measures of the Spanish test, measures that, when once taken into account, do not change qualitatively the results discussed so far (see in Section 4.1). Finally, we estimate the dimension of the under-counting of deaths among the elderly in the different countries and give estimations for the overall lethality of the virus in Section 4.2. We relegate all the details concerning the databases and dates used in the data-analysis for the Section 2" to methods, if needed.

6. PLOS authors have the option to publish the peer review history of their article (what does this mean?). If published, this will include your full peer review and any attached files.

Reviewer #1: No

Reviewer #2: No

---

## [Author Response · Author response to Decision Letter 0]

20 Nov 2020

Reviewer #1: Review of "A scaling approach to estimate the COVID-19 infection fatality ratio from incomplete data"

This paper studies age-patterns of death due to COVID-19. The goal is to try to identify regularities across countries; these regularities can then be used as the basis for predicting unobserved quantities in some countries based on the experience of other countries where more complete data are available. For example, a high-quality sero-survey is available in Spain, and the paper tries to develop methods can can be used to project information from the Spanish sero-survey onto data from other settings, where no sero-survey is available.

This is an interesting and important topic, and it seems clear that the analysis described in the paper is based on a thoughtful consideration of many aspects of this problem. However, the paper currently reads like a very early draft, to the point where it is quite difficult to understand what is happening in the analysis. There are some minor issues with the English language, but that is not the primary concern -- rather, the organization, lack of key detail, and inclusion of what seem like extraneous details, are the primary challenges I had in understanding the argument here. I think it's possible that the ideas here could be the basis of a paper that is publishable, but in my opinion, this draft needs major revisions before it is ready to really be evaluated.

We thank again the referee for considering the paper interesting and describing it as “based on a thoughtful consideration of many aspects of this problem”. In the new version of the manuscript, we tried to improve the writting and the clearity of the methods and results.

Major comments:

* pg 2, 2nd paragraph - I find the assertion that mortality due to non-COVID causes has decreased to be surprising. Is that really known definitively? For many causes, it's plausible that death rates have increased: routine medical care has slowed down, lots of people are likely suffering from depression and other mental health challenges because of the economic and social disruption. If this claim is made, there should be citations or evidence provided for it. Otherwise, it seems like speculation

We explained better this assertion and added some new references, as suggested by the referee.

* pg 2, I would find it helpful if the information about the data sources was summarized in a table

We added a new supplemental table (Table S1), detailing all the data sources used for each figure in the paper. We also rewrote the Materials and Methods section to be more explicit about this.

* pg 3, I don't think that eqn (1) is the mortality rate (or death rate) as conventionally used in demography/epidemiology, because the denominator does not appear to have anything related to time. (Conventionally, a mortality rate has in its denominator a measure of exposure, ie people X time). For example, unless I'm reading eqn (1) incorrectly, it says that a population observed for twice as long -- but with no change in mortality conditions -- would have twice as high a value of \\had{D}_\\alpha. This may not cause problems for the rest of the analysis, but anyone used to working with demographic/epidemiological data would likely find it confusing to call it a 'mortality rate'. So, assuming the lack of time in the denominator is intentional, I suggest calling this something like 'cumulative fraction dead' or, anyway, something other than 'mortality rate'

We agree with the referee that using the term mortality rate was a careless term here, so we removed the sentence below the equation mentioning the mortality rate. We also removed this term later in the text and used the term “normalized cumulative deaths” instead.

* pg 3, 'expected error of an un-normalized histogram' - what does this mean? Where does this come from? There is no citation or explanation. I don't think I've ever seen this phrase in an epidemiological paper before. This is likely related to my confusion about what is being treated asymptotically (see 'additional comments')

We added a footnote (footnote no. 2) in Page 4 explaining in detail this statement.

* pg 4, is r_\\alpha not net of population size? It looks like r_\\alpha = \\frac{I_\\alpha}{x_\\alpha I}; that seems counter-intuitive; the first time I read this, I assumed the natural quantity to look at would be r_\\alpha = \\frac{I_\\alpha N}{x_alpha I}. So I would find it helpful to motivate and explain in more detail what r_\\alpha is

We included a larger explanation now. The advantage of using our definition of r_alpha is that it allows us to move from real to uniform exposure of the virus just by tuning this parameter to 1.

* Fig 2 - how is 'excess deaths' defined in the calculations shown in this figure? It looks like a relative quantity, but I'm guessing - there's no detail provided about this. Pg 5 says 'we give these details ... in the Methods and Dataset section' -- but I don't see any section with that title?

We thank the referee for pointing this error. We now define the ratio of under-counting in the Methods (new Eq. (1)) in a Section called “Under-reporting of deaths” and refer to it in the discussion about the under-counting. We also remove the term excess deaths because it’s misleading. We also modified the labels in the Fig. 2 accordingly.

* pg 5 - "'apparent' fatality (what we perceive from the daily news)" - this sounds like a useful point, but I did not really understand how \\hat{f}_\\alpha is what is shown in the news. I suggest explaining this in greater detail

We reformulated the explanation in the text.

* Pg 6-7 / Fig 3 - How exactly was this fit performed? Maximum likelihood? Least squares? Also, I wasn't convinced that the efforts to assess model fit took into account the complexities of the log-log scale, which can make assessing fit difficult. On that topic, I find that this paper has a useful discussion:

Clauset, Aaron, Cosma Rohilla Shalizi, and Mark EJ Newman. "Power-law distributions in empirical data." SIAM review 51.4 (2009): 661-703.

(The authors may have done this kind of thing - but it's not described in the paper.)

The fit was done using the least squares methods. I agree with the referee that log-log fits are tricky, but my fit is log-linear, that is, it’s a purely exponential growth, so I do not think the reference suggested applies here. Neither does the worries, least-squares method is a standard method to do this kind of fit. Nevertheless, we followed the referee’s suggestion and mentioned the fitting method in footnote 5 in page 7, and explicitely wrote that we are doing a linear regresion of the logarithm of the normalized number of deaths.

Additional comments:

Note: there are many typos, missing words, and other writing issues throughout the paper. I will note some of them here, but I eventually stopped keeping track of these. The paper needs a lot of editing.

We revised the manuscript carefully and tried to solve this important issue.

- abstract: I find it a little confusing to refer to the "growth" of the mortality rate with age; we don't usually talk about death rates 'growing'. I would suggest "increase" instead of "growth"

I thank the author for the suggestion. I now modified the terminology all over the text.

- abstract: 'validate' seems like a strong word for the analysis here; perhaps 'evaluate'?

We modified the text to soften the statement, as suggested.

- abstract: I believe the IFR is the 'infection fatality ratio', not 'fatality infection ratio'

The error was corrected.

* pg 2, 1st paragraph - infection fatality ratio is defined, but not case fatality ratio; I suggest defining CFR the same way IFR is defined, in parenthesis after first using the term

I followed the referee’s suggestion and included the term CFR in the text.

* pg 2, 1st paragraph - 'cruiser' is not the right word. "Cruise ship", maybe?

The error is now corrected.

* pg 2, 1st paragraph - it's not clear to me what is meant by "given how elusive the detection of the virus is". Is this referring to a lack of testing? Or the difficulty of devising accurate tests?

The sentence was modified to “given the difficulty to systematically identify all the mild and asymptomatic infections”.

* pg 2, 3rd paragraph -- 'dimension' does not seem like the right word here. Maybe the 'extent' of under-counting of deaths?

We modified the word, as suggested.

* pg 3, 'Asymptotically' - I can't understand what 'asymptotically' refers to without setting up what is getting arbitrarily big. Is it the sample size? The amount of time elapsed? Is either of those scenarios important here for some reason?

Asymtotically refers here to a large number of infections, so that fluctuations small. We have written it explicitely in the text, and added a footnote in Page 4 explaining the derivation of the formulas.

* pg 3, 'assignation' - I don't think this is the right word here

We changed assignation→ assignment

* Fig 1, caption - what is UIFR?

UIFR is defined in the text (just below Eq. (6)), as the IFR under the assumption of uniform distribution of the infections. Yet, we included also a clarification in the caption of Fig. 1, to remind the reader the difference between the real IFR and the IFR computed under the assumption of a uniform attack rate (the UIFR).

Reviewer #2: 

We thank the referee for his/her positive evaluation of the statistical significance of the analysis and the results discussion.

When you state the sentence "The actual degree of under-counting for both measures is unknown and most likely country dependent, which results in largely irreconcilable case fatality ratios all over the world." you should back it with a citation (Chirico F, Nucera G, Magnavita N. Estimating case fatality ratio during COVID-19 epidemics: Pitfalls and alternatives. J Infect Dev Ctries. 2020;14(5):438-439. Published 2020 May 31. doi:10.3855/jidc.12787).

We included this reference (and another 3) as suggested by the referee.

In Introduction you should move this "We define all our variables in Section 2.2. We establish a

direct correspondence between the mortality rates in patients below 70 years old (where we argue the official counting

is more accurate) published in different countries around the world (but mostly in Europe) in Section 3.2. This good

correspondence allows us to make predictions about the degree of spread of the virus in different populations, or the

global IFR of a country, as compared to another one. We also observe that the collapse of the mortality rate with age

in different countries is compatible with a pure exponential growth of the IFR with age (assuming a uniform attack

rate). The scale of total infections is then consistently fixed from the rate of immunity obtained via blood tests of a

statistical sampling of the citizens Spain in Section 3.3 (and compared to seroprevalence tests in Geneva, Switzerland,

and New York City, United States). This scale allows us to compute the IFR as function of age and the number of

current infections in each country that are given in Table 2. In addition, we estimate the probability of being detected as

official case, needing hospitalization and intensive care (if infected) as function of age in Spain in Section 3.4. All these

rates are obtained under the assumption of a uniform attack rate, an assumption that seems fairly reasonable seeing the immunity measures of the Spanish test, measures that, when once taken into account, do not change qualitatively the results discussed so far (see in Section 4.1). Finally, we estimate the dimension of the under-counting of deaths among the elderly in the different countries and give estimations for the overall lethality of the virus in Section 4.2. We relegate all the details concerning the databases and dates used in the data-analysis for the Section 2" to methods, if needed.

We followed the referee’s suggestion, and moved the paragraph.

---

## [Decision Letter · Decision Letter 1]

25 Jan 2021

PONE-D-20-17835R1

A scaling approach to estimate the age-dependent COVID-19 infection fatality ratio from incomplete data

PLOS ONE

Dear Dr. Seoane,

Thank you for submitting your manuscript to PLOS ONE. After careful consideration, we feel that it has merit but does not fully meet PLOS ONE’s publication criteria as it currently stands. Therefore, we invite you to submit a revised version of the manuscript that addresses the points raised during the review process.

We look forward to receiving your revised manuscript.

Kind regards,

Wen-Jun Tu

Academic Editor

PLOS ONE

Additional Editor Comments (if provided):

1. In order to provide a more complete information to our readers on the topic, we would like to emphasize the importance to cross referencing very recent material on the same topic published in "PLoS ONE ". Therefore, it would be highly appreciated if you would check the contents published in the last two years of "PLoS ONE" (https://journals.plos.org/plosone/) and add all material relevant to your article to the reference list.

2. Add "Clinical Features and Short-term Outcomes of 102 Patients with Corona Virus Disease 2019 in Wuhan, China. Clinical Infectious Diseases, 71(15):748-755" in the revision text

Reviewers' comments:

Reviewer's Responses to Questions

**Comments to the Author**

1. If the authors have adequately addressed your comments raised in a previous round of review and you feel that this manuscript is now acceptable for publication, you may indicate that here to bypass the “Comments to the Author” section, enter your conflict of interest statement in the “Confidential to Editor” section, and submit your "Accept" recommendation.

Reviewer #2: All comments have been addressed

2. Is the manuscript technically sound, and do the data support the conclusions?

Reviewer #2: Yes

3. Has the statistical analysis been performed appropriately and rigorously? 

Reviewer #2: Yes

4. Have the authors made all data underlying the findings in their manuscript fully available?

Reviewer #2: Yes

5. Is the manuscript presented in an intelligible fashion and written in standard English?

Reviewer #2: Yes

6. Review Comments to the Author

Reviewer #2: (No Response)

7. PLOS authors have the option to publish the peer review history of their article (what does this mean?). If published, this will include your full peer review and any attached files.

Reviewer #2: **Yes: **Prof Francesco Chirico

---

## [Author Response · Author response to Decision Letter 1]

25 Jan 2021

I thank the referees for their careful read of the manuscript and their agree to publication in PLoS ONE.

---

## [Editor Report · Decision Letter 2]

27 Jan 2021

A scaling approach to estimate the age-dependent COVID-19 infection fatality ratio from incomplete data

PONE-D-20-17835R2

Dear Dr. Seoane,

We’re pleased to inform you that your manuscript has been judged scientifically suitable for publication and will be formally accepted for publication once it meets all outstanding technical requirements.

Kind regards,

Wen-Jun Tu

Academic Editor

PLOS ONE
---

## [Editor Report · Acceptance letter]

9 Feb 2021

PONE-D-20-17835R2 

A scaling approach to estimate the age-dependent COVID-19 infection fatality ratio from incomplete data 

Dear Dr. Seoane:

I'm pleased to inform you that your manuscript has been deemed suitable for publication in PLOS ONE. Congratulations! Your manuscript is now with our production department. 

Kind regards, 

on behalf of

Dr. Wen-Jun Tu 

Academic Editor

PLOS ONE